# A Two-Step Rule-Extraction Technique for a CNN

**Guido Bologna [1,2,\*]**  **and Silvio Fossati [3]**

[1] Department of Computer Science, University of Applied Sciences and Arts of Western Switzerland, Rue de la Prairie 4, 1202 Geneva, Switzerland

[2] Department of Computer Science, University of Geneva, Route de Drize 7, 1227 Carouge, Switzerland

[3] Independent Researcher, 1202 Geneve, Switzerland; fossati.silvio@hotmail.com

[\*] Correspondence: guido.bologna@hesge.ch; Tel.: +41-22-5462551

**Abstract:** The explanation of the decisions provided by a model are crucial in a domain such as medical diagnosis. With the advent of deep learning, it is very important to explain why a classification is reached by a model. This work tackles the transparency problem of convolutional neural networks(CNNs). We propose to generate propositional rules from CNNs, because they are intuitive to the way humans reason. Our method considers that a CNN is the union of two subnetworks: a multi-layer erceptron (MLP) in the fully connected layers; and a subnetwork including several 2D convolutional layers and max-pooling layers. Rule extraction exhibits two main steps, with each step generating rules from each subnetwork of the CNN. In practice, we approximate the two subnetworks by two particular MLP models that makes it possible to generate propositional rules. We performed the experiments with two datasets involving images: MNISTdigit recognition; and skin-cancer diagnosis. With high fidelity, the extracted rules designated the location of discriminant pixels, as well as the conditions that had to be met to achieve the classification. We illustrated several examples of rules by their centroids and their discriminant pixels.

**Keywords:** CNN; model explanation; rule extraction; unordered rules

## 1. Introduction

More than 30 years ago multi-layer perceptrons (MLPs) started to learn numerous classification problems, especially with shallow architectures [1]. A major drawback with MLPs is the difficulty in explaining their responses. The transparency of bio-inspired models is an important research topic, as transparency is essential with respect to the European General Data Protection Regulation (GDPR). Specifically, a right of explanation is crucial, because when a mathematical model decides, an individual has the right to ask for a valid explanation. An intuitive approach to revealing the reasoning behind MLPs is the use of propositional rules [2]. The first taxonomy depicting the general characteristics of all rule-extraction techniques from connectionist models was introduced in [3]. With the advent of deep learning, MLP architectures have become even more complex and their opacity has increased.

Deep models such as convolutional neural networks (CNNs) contributed to the significant progress accomplished over the past ten years in areas such as artificial vision, natural language processing and speech recognition. In image classification, many techniques that explain CNN decisions learn an interpretable model in the local region close to an input sample [4]. The main drawback of local algorithms is their difficulty to apprehend a phenomenon in its entirety. As a result, the state of the art underlines a shortage of global methods for extracting symbolic rules from CNNs. For instance, Guidotti et al. presented a survey on black-box models with its "explanators" [5]. In addition, Adadi and Berrada proposed an overview of explainable artificial intelligence (XAI), including neural networks [6]. Finally, it should be noted that several algorithms aiming at explaining

CNN decisions in image classification visualize areas that are mainly relevant for the outcome [7]. Nevertheless, as stated by Rudin [8]: "it does not explain anything except where the network is looking". Moreover, saliency maps could be very similar for several different classes.

In this work, we propose a two-step technique that generates propositional rules from CNNs trained with images comprising convolutional layers, max-pooling layers and fully connected layers (also denoted as dense layers). Our technique is global, as we apply it to all samples of the input space. In the first step, we approximate the fully connected layers at the top of the CNN by a special MLP that makes it possible to generate propositional rules. The approximator of the dense layers is denoted as VDIMLP (virtual discretized interpretable multi-layer perceptron). This model is a specific DIMLP model [9] that approximates the dense subnetwork of the original CNN to any desired precision.

Rule extraction in DIMLPs is performed by determining axis-parallel discriminative hyperplanes [9–12]. Each propositional rule $R_i$ generated from VDIMLP is then used to define a new binary classification problem with classes: "rule covered"; "rule non-covered". At this stage, the antecedents of the extracted rules represent the responses of convolutional filters. Then, an interpretable convolutional subnetwork that approximates the convolutional layers of the original CNN is introduced to generate propositional rules with respect to the new binary problem. This interpretable approximator denoted as DICON (discretized interpretable convolutional network) allows us to obtain propositional rules that depends on data inputs. We perform the experiments on two datasets: MNIST digit recognition; and skin-cancer diagnosis. The generated rules clearly indicate the location of the important pixels, as well as the conditions that must be met to achieve the classification.

This article extends the work proposed in [13]. Here, the rule-extraction technique has been formalized with two main steps: rule extraction in the dense layers; and rule extraction in the convolutional layers, which was not carried out in [13]. The main advantage of introducing VDIMLP in the CNN is that the hard problem of rule extraction is divided into many rule-extraction subproblems that are easier to solve. These subproblems depend on the rules generated by VDIMLP in the upper layers. Since these rules are unordered, each rule is itself a classifier that allows us to decompose the original classification problem into many binary classification problems. Each binary classification problem is approximated by a DICON in the lower layers of the CNN, which is only a small part of it. Practically, rule extraction from DICONs run in parallel, which is also a clear advantage. In addition, we apply the current approach to the skin-cancer problem, which is illustrated with several examples. Finally, we perform a comparison with decision trees and DIMLP ensembles; it shows that our new approach is more accurate. In another work we generated propositional rules from a CNN that was trained on sentiment analysis data [14]. The architecture included a unique convolutional layer based on wide convolution. Here, we achieve rule extraction to more complicated CNN architectures that classify images with an arbitrary number of convolutional layers. In the following paragraphs, Section 2 illustrates a brief state of the art with respect to interpretability of deep architectures, Section 3 describes the models and the rule-extraction technique, Section 4 presents the experimental results, followed by the conclusion.

## 2. Rule Extraction from Deep Networks

Our purpose is to determine propositional rules, since they are very close to the human reasoning. Let us first define an antecedent $A_i$ of a propositional rule as: $a_i < t_i$, or $a_i \geq t_i$; with $a_i$ an input variable and $t_i$ a constant. A propositional rule with $n$ antecedents is: "if $A_1$ and ... and $A_n$ then conclusion". For data classification, "conclusion" is a class that can be characterized with the values of the antecedents. Golea proved that the problem of rule extraction from MLPs is NP-hard [15]. Andrews et al. [3] introduced a taxonomy to characterize all rule-extraction techniques from neural networks. Basically, they distinguished three main categories of methods, namely: pedagogical; decompositional; and eclectic. Pedagogical techniques use a transparent model, such as decision trees (DTs) [16] to learn input/output associations, without taking into account weight values.

Decompositional methods generates propositional rules by analyzing the weight values. However, the majority of algorithms in this category presents exponential algorithmic complexity. To alleviate this difficulty, a pruning step is helpful to reduce the number of weights. Finally, eclectic techniques are both pedagogical and decompositional. It is worth noting that the simplest pedagogical approach of adapting a DT to a deep neural network generally gives unsatisfactory results in terms of accuracy and fidelity [17].

Bologna and Hayashi extracted propositional rules from deep DIMLPs trained with several stacked auto-encoders [18,19]. Similarly, Zilke tackled the same problem [20,21], but with a technique that first generate the rules between two successive layers. In the end, the rules involving the input layer with the output layer were produced by transitivity. With deep beliefs networks (DBNs), Boolean formulas were extracted from two successive layers with a certain level of confidence [22], then rules were generated by chaining down the Boolean formulas from the output layer to the input layer. In [23], Hayashi transferred the last layer of DBN weights into MLPs with a unique hidden layer; then, rules were produced with the Re-RX algorithm [24]. Finally, Nguyen et al. proposed an exact transformation of deep MLPs into multivariate decision trees [25]. The generated rules comprise in the antecedents' linear combinations of the inputs that might be difficult to understand, especially for high-dimensional problems.

Several authors proposed local methods to explain the decisions taken by CNNs. As an important approach, Ribeiro et al. presented LIME (locally interpretable model agnostic explanations). The goal of this technique is to learn a transparent model in the neighborhood of an input sample [4]. The authors of LIME generated explanations with the use of linear models. LORE (local rule based explanations) represents another representative technique; it learns a DT on a neighborhood generated through a genetic algorithm [5]. In the end, rules are generated from the DT. Generally, pedagogical rule-extraction techniques could be applied to deep architectures such as CNNs. Among the rare examples, Frosst and Hinton proposed to use a CNN to train a particular DT that imitates the input-output associations found out by the CNN [26]. Nevertheless, the authors stated that the DT did not explain the network logic, clearly. In [27] a DT determined with respect to images, which filters were activated by objects parts and how much they contributed to the final classification.

To explain CNN classifications, different approaches characterized the areas of the image that are primarily relevant to the output [7]. For instance, Zeiler and Fergus described DeconvNet [28]. Specifically, strong activations are propagated backward to determine parts of the image causing these activations. Mahendran and Vedaldi presented an optimization technique based on image priors to invert a CNN [29]. With such an approach, it was possible to visualize the information represented at each layer. Layer-wise relevance propagation (LRP) is a technique that determine heatmaps of relevant areas contributing to the final classification [30]. Interestingly, the authors noted that the context is important in the recognition process of objects in images. With the use of medical images, Holzinger et al. characterized the internal structure of CNNs by replacing many convolutional layers by AM-FM components (amplitude modulation-frequency modulation) and by retraining the upper network layers [2].

Features importance is a measure aiming at determining the importance of inputs [5]. It brings to light specific properties of the model, without necessitating an overall understanding of it. Likewise, sensitivity analysis determines the importance with respect to the variation of the inputs. With propositional rules, classes are characterized by rule antecedent values. This is much more precise than simply determining the relevant image subregions emphasized by heat maps. With those maps or with the list of important variables, the way in which discrimination between different classes is carried out is undetermined [8]. The main difference between our method and the most recent techniques described in [5,6,31] aiming at generating decision trees or symbolic rules from CNNs is that our approach is global and the majority of the others are local. In addition, our technique first generates abstract rules from the top layers and then produces comprehensible rules at the low level.

## 3. Models

### 3.1. MLP and VDIMLP

In feed-forward neural network such as MLPs, neurons are arranged into successive layers. We define $x^{(0)}$ as the input layer of an MLP; it is, therefore, a vector whose dimensionality depends on the dataset to be learned. For layer $l + 1$, the activation values $x^{(l+1)}$ of the neurons are given by

$$x^{(l+1)} = Act(W^l x^{(l)} + b^{(l)}); \tag{1}$$

with $W^l$ a matrix consisting of weights between two successive layers $l$ and $l + 1$, vector $b^{(l)}$ representing the bias vector and with *Act* corresponding to an activation function which is very often a sigmoid $\sigma(x)$ given as:

$$\sigma(x) = \frac{1}{1 + \exp(-x)}. \tag{2}$$

A discretized interpretable multi-layer perceptron (DIMLP) is an MLP, but with two differences. The first is $W^0$, which is a diagonal matrix. The second is the activation function applied to $x^{(1)}$, which is a staircase function $S(x)$ that approximates with $\Theta$ stairs an arbitrary activation function $G(x)$ on a compact interval:

$$S(x) = A_{min}, \quad \text{if } x \leq A_{min}; \tag{3}$$

$A_{min}$ represents the abscissa of the first stair. By default $A_{min} = -5$.

$$S(x) = A_{max}, \quad \text{if } x \geq A_{max}; \tag{4}$$

$A_{max}$ represents the abscissa of the last stair. By default $A_{max} = 5$. Between $A_{min}$ and $A_{max}$, $S(x)$ is:

$$S(x) = G(A_{min} + \left[\Theta \frac{x - A_{min}}{A_{max} - A_{min}}\right] (\frac{A_{max} - A_{min}}{\Theta})). \tag{5}$$

Since $W^0$ is diagonal the input space is split into hyper-rectangles representing propositional rules. Then, the basic trick in the rule-extraction algorithm is to precisely determine the location of axis-parallel hyperplanes, thanks to the staircase activation function [11]. Please note that these hyperplanes are effective or not, depending on the weight values of $W^l$, with $l > 0$. The number of stairs in the staircase activation function determines the number of hyperplanes. At this point, a decision tree is built with respect to the training set and in perfect agreement with neural network responses. When the tree is completed, each path from the root to a leaf represents a propositional unordered rule. Finally, a greedy algorithm progressively removes antecedents and rules. More details on the rule-extraction algorithm can be found in [11].

If we replace the activation function applied to $x^{(1)}$ by a staircase function that approximates the identity function ($Id(x) = x$), we obtain a DIMLP that approximates an MLP. In this case, the diagonal matrix $W^{(0)}$ is an identity matrix, while the bias vector has values equal to zero. In addition, the greater the number of steps in the staircase activation function the better the approximation. Virtually, for a given MLP there will exist a DIMLP that approximates the MLP to an arbitrary precision. As shown in Figure 1, we introduce the VDIMLP model. It designates a virtual DIMLP that can approximate any (standard) MLP composed of an input layer, an arbitrary number of hidden layers and an output layer.

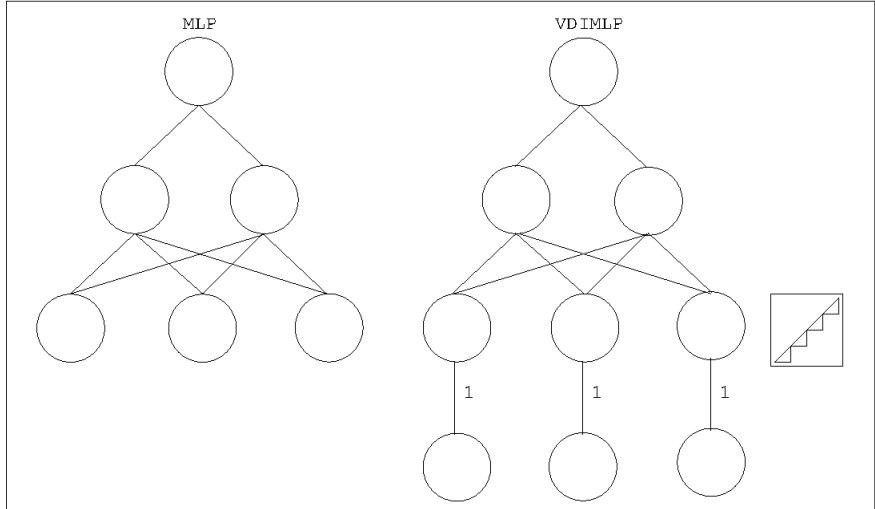

**Figure 1.** The concept of multi-layer erceptron (MLP) approximation by virtual discretized interpretable multi-layer perceptron (VDIMLP) networks, with MLP as a subnetwork of VDIMLP. The weight values of the two networks are equal from the first hidden layer of VDIMLP. Between the input layer and the first hidden layer, VDIMLP approximates the Identity function with a staircase activation function. This approximation allows us to generate propositional rules.

### 3.2. Convolutional Neural Networks

Convolution is at the heart of CNNs. Here we consider the two-dimensional convolution operator applied to images. Therefore, given a two-dimensional kernel $w_{pq}$ of size $P \times Q$ and a data matrix of elements $m_{ab}$, the calculation of an element $c_{ij}$ of the convolutional layer is

$$c_{ij} = f(\sum_{p}^{P} \sum_{q}^{Q} w_{pq} m_{i-p, j-q} + b_{pq}); \tag{6}$$

with $f$ a transfer function and $b_{pq}$ the bias. As a transfer function we use the ReLU (rectified linear unit):

$$ReLU(x) = Max(0, x). \tag{7}$$

Another important operator in CNNs is max-pooling. It reduces the size of a matrix by applying a "max" operator over non-overlapping regions. For instance, Figure 2 illustrates how max-pooling is applied to a $4 \times 4$ matrix with respect to $2 \times 2$ non-overlapping regions.

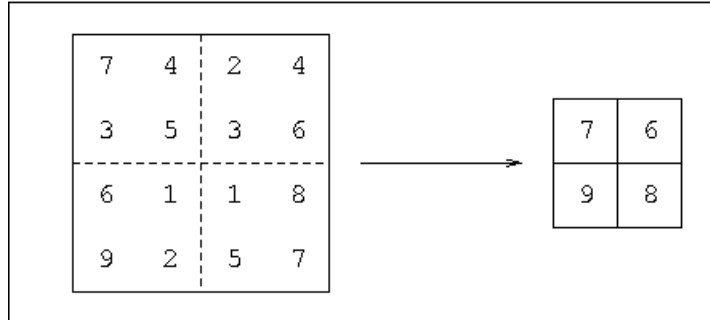

**Figure 2.** The max-pooling operator applied to a $4 \times 4$ matrix with four non-overlapping regions of size $2 \times 2$.

Typically, after the convolutional layers and the max-pooling layers lie the fully connected layers. The lower layers act like feature extractors, which are then processed by the dense layers representing

an MLP. Our rule-extraction technique first generates propositional rules from the MLP subnetwork and then rules are extracted from the lower layers.

### 3.3. Rule Extraction from CNNs

For clarity, let us describe the CNNs with respect to different types of layers. Here, we consider a two-dimensional input layer that is denoted as $X^{(0)}$. In addition, we focus on CNNs including layers $X^{(0)}-\{C \cup M\}^+-D^+$; with $C$ representing convolutional layers, $M$ standing for max-pooling layers and $D$ representing fully connected layers. Symbol "+" designates one or more occurrences of a given layer.

We define the discretized interpretable convolutional network (DICON) as an approximator of layers $X^{(0)}-\{C \cup M\}^+$ included in a CNN. DICON has the following layers: $X^{(0)}-X^{(1)}-\{C \cup M\}^+$. In practice, we add a new layer $X^{(1)}$ with the same number of neurons as $X^{(0)}$. The values in $X^{(1)}$ are calculated by applying a convolution kernel of size $1 \times 1$ with a value equal to one, without bias. In addition, the activation function applied to $X^{(1)}$ is a staircase function that approximates the identity function. Please note that this Identity approximation is similar in VDIMLPs, but with a two-dimensional input layer.

From a CNN we extract unordered rules, which implies that each rule is a single piece of knowledge that we can examine without looking at the others. On the contrary, in a ruleset composed of ordered rules the "else" statement is implicit between two successive rules. Hence, a long list of ordered rules has many implicit antecedents that makes the interpretation difficult. Finally, with ordered rules a given sample activates a unique rule, while for unordered rules it can activate more than one rule.

We carry out rule extraction in two steps:

- rule extraction from the fully connected layers ($D^+$) with VDIMLP as an approximator;
- rule extraction from the input layer to the fully connected layers ($X^{(0)}-\{C \cup M\}^+$) with DICON as an approximator.

The rules generated by VDIMLP bring into play in the antecedents the filters responses. This is of interest to characterize the way filters interact to classify data, but it is not yet sufficient to explain CNN responses, relative to the neurons in the input layer. Hence, in the second step of the proposed rule-extraction technique we make the connection between a ruleset generated from VDIMLP and the input layer.

Each rule generated from VDIMLP defines a new binary classification problem that a DICON can process. Thus, for each binary classification problem the positive class corresponds to the samples that activate the VDIMLP rule. On the contrary, the negative class represents the training samples that do not trigger the VDIMLP rule. A clear advantage of this decomposition is that we can run in parallel all rule extractions for new binary classification problems.

With respect to each new binary classification problem, DICON requires two supplemental layers. The first added layer determines whether a rule antecedent of a rule generated from VDIMLP is true or not. The second added layer performs a logical "and" of the antecedents. Therefore, rules are extracted from a neural network containing a DICON and two dense layers, defined as DICON-$D^2$. Figure 3 presents the coding of a rule in the DICON top layers (layers $D^2$).

At the bottom, neurons $f_1$, $f_2$, and $f_3$ represent the antecedents of a rule extracted from VDIMLP. The intermediate layer with a step activation function detects whether any antecedent is true or false. It is worth noting that the weight values of the first bias neuron correspond to the thresholds of the antecedents, in absolute values. The output neuron at the top is activated with a value equal to one (meaning that the rule at the bottom is true), if and only if all the neurons in the middle have an activation equal to one.

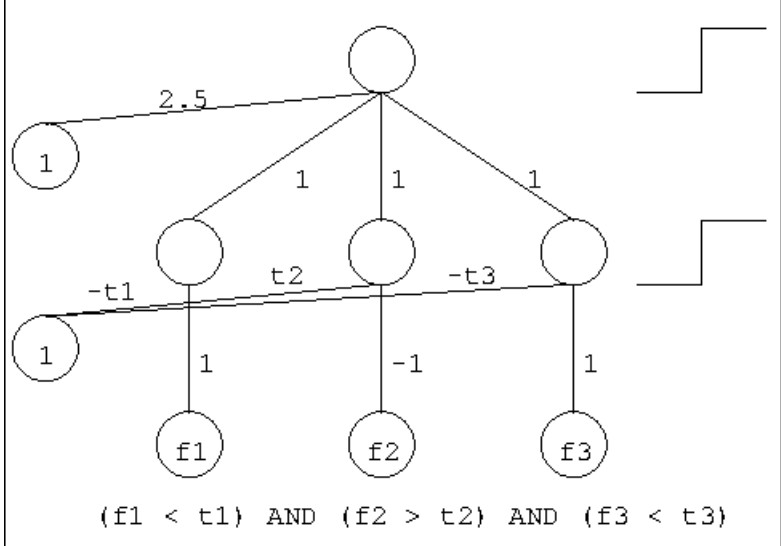

**Figure 3.** Coding of a propositional rule with three antecedents. The first layer of weights specifies the antecedents, while the second layer achieves the logical "and" of the antecedents. The activation function of the neurons is a step function.

## 4. Results

We illustrate the experiments with three classification problems. The first experiment is related to MLPs applied to a benchmark dataset, which is called the thyroid dataset. The purpose is to show that VDIMLP approximates an MLP adequately at any hidden layer. In the other two experiments, CNNs learn from image-based datasets. Specifically, we use the well-known MNIST dataset of hand-written digits and skin-cancer images. Note that the input features of the thyroid classification problem are not related to images or temporal series. Thus, at priori it is more appropriate to learn this dataset with a model such as an MLP, rather than a CNN. All the datasets are in the public domain. They are available on the following Web pages:

- Thyroid dataset: UCI archive: http://archive.ics.uci.edu/ml/
- MNIST dataset: http://yann.lecun.com/exdb/mnist/
- Skin-Cancer dataset: https://www.kaggle.com/fanconic/skin-cancer-malignant-vs-benign

In the experiments, the accuracy measure is crucial to evaluate the models and their extracted propositional rules. Generally, for a given dataset $D$ the accuracy of classifier $C$ $A_C$ is:

$$A_C(D) = \frac{\#correct(D)}{\#correct(D) + \#wrong(D)};$$

(8)

with $\#correct(D)$ indicating the number of samples correctly classified and $\#wrong(D)$ those incorrectly classified. If the classifier $C$ is an ordered ruleset, each sample of $D$ activates a unique rule, which belongs to a given class. Hence, Equation (8) is still valid.

The use of an unordered ruleset is somewhat more complicated when a sample activate more than one rule, because of the potential conflicts between different classes. In such a situation, if the neural network response corresponds to a class that is present among the classes represented by the activated rules, the classification is considered correct; otherwise, it is wrong.

### 4.1. Thyroid Dataset

The Thyroid dataset consists of 3772 samples for the training set and 3428 samples for the testing set, respectively. In addition, the number of inputs is 21 and the number of classes is three. We define MLPs with two hidden layers and demonstrate that propositional rules can be produced at any layer.



Although a single hidden layer is sufficient to learn any classification problem, we defined two hidden layers, to illustrate how the number of generated rules varies at each hidden layer. We trained ten MLPs with ten neurons in the first hidden layer and five neurons in the second. The training phase was stopped after learning 99.5% of the training set. Table 1 depicts averaged results over ten trials. Specifically, the rows indicate MLP, VDIMLP approximating MLP from the input layer to the output layer (VDIMLP0), VDIMLP approximating MLP from the first hidden layer (VDIMLP1) and VDIMLP approximating MLP from the second hidden layer (VDIMLP2). In addition, we illustrate from left to right average values of:

- predictive accuracy on the testing set;
- fidelity, which is the degree of matching between generated rules and the neural network on the testing set;
- predictive accuracy of the rules (testing set);
- predictive accuracy of the rules when rules and network agree (testing set);
- number of extracted rules;
- average number of rule antecedents.

**Table 1.** Average results on the thyroid dataset. From left to right: predictive accuracy, fidelity, predictive accuracy of the rules, predictive accuracy of the rules when rules and network agree, number of rules, number of antecedents. Numbers in brackets designate standard deviations.

|          | Tst. Acc.  | Fid.      | Rul. Acc. (1) | Rul. Acc. (2) | #Rul.       | #Ant.     |
|----------|------------|-----------|---------------|---------------|-------------|-----------|
| MLP      | 97.5 (0.3) | -         | -             | -             | -           | -         |
| VDIMLP0  | 97.3 (0.2) | 98.5 (0.3)| 97.9 (0.3)    | 98.3 (0.2)    | 38.3 (6.5)  | 4.3 (0.2) |
| VDIMLP1  | 97.5 (0.3) | 98.3 (0.3)| 96.9 (0.3)    | 98.1 (0.3)    | 79.7 (11.7) | 4.1 (0.2) |
| VDIMLP2  | 97.5 (0.3) | 99.7 (0.1)| 97.4 (0.4)    | 97.6 (0.3)    | 9.8 (1.4)   | 2.6 (0.2) |

Average predictive accuracy provided by MLPs and VDIMLPs is very similar. With respect to the generated rules, average predictive accuracy is a bit higher for VDIMLP0 (97.9% versus 97.5%) and a bit lower for VDIMLP1 (96.9% versus 97.5%). The highest average fidelity and the lowest average number of rules is reached by VDIMLP2 (99.7% and 9.8). Nevertheless, the rules by VDIMLP2 are not easy to understand, since the antecedents correspond to the activation values of the MLP second hidden layer. Please note that the most interpretable model is VDIMLP0, since the antecedents represent neurons of the MLP input layer.

For comparison purposes, Table 2 depicts average results based on ten trials, obtained by DIMLP ensembles trained by Bagging [32], single DIMLP networks and C4.5 decision trees [33]. DIMLP ensembles consisted of 25 networks with a "default" architecture consisting of a single hidden layer comprising the same number of input layer neurons. For single DIMLPs and C4.5 DTs, the average predictive accuracy of the rules was very close (99.3% versus 99.4%). Not surprisingly, more rules were extracted from the DIMLPs than C4.5 DTs (16.5 versus 7.0), since DIMLP rules are not ordered.

**Table 2.** Average results obtained by discretized interpretable multi-layer perceptron (DIMLP) ensembles, single DIMLP networks and C4.5 decision trees.

|           | Tst. Acc.  | Fid.      | Rul. Acc. (1) | Rul. Acc. (2) | #Rul.      | #Ant.     |
|-----------|------------|-----------|---------------|---------------|------------|-----------|
| DIMLP-ens | 98.6 (0.1) | 99.5 (0.2)| 98.7 (0.2)    | 99.0 (0.1)    | 24.5 (6.1) | 3.5 (0.2) |
| DIMLP [34]| -          | -         | 99.3 (0.0)    | -             | 16.5 (-)   | 3.4 (-)   |
| C4.5 [34] | 99.4 (0.0) | -         | 99.4 (0.0)    | -             | 7.0 (0.0)  | 2.0 (0.0) |

### 4.2. MNIST Dataset

The MNIST dataset is about hand-written digit classification [35]. Digits between zero and nine are represented by $28 \times 28$ (= 784) inputs with normalized gray levels. The training and testing sets of

this classification problem comprise 60,000 and 10,000 samples, respectively. Note also that the last 10,000 samples of the training set were used as a tuning set for early stopping [36].

Many authors tackled this benchmark problem with increasingly complex architectures [35]. Our purpose is to illustrate rule extraction with a kind of "default" architecture that provides good accuracy. The training phase was performed with Lasagne libraries, version 0.2 [37]. In practice, we defined a CNN architecture comprising the following layers:

- an input layer;
- a convolutional layer with 32 kernels of size $5 \times 5$ and ReLU activation function, the stride parameter being equal to 1;
- a max-pooling layer, the stride parameter being equal to 2;
- a convolutional layer with 32 kernels of size $5 \times 5$ and ReLU activation function, the stride parameter being equal to 1;
- a max-pooling layer, the stride parameter being equal to 2;
- a fully connected layer of 256 neurons;
- an output layer of 10 neurons.

A Lasagne script that defines this CNN architecture is available on https://lasagne.readthedocs.io/en/latest/user/tutorial.html. We just modified the second convolutional layer to obtain a lower number of parameters. Without this change the number of kernels in this layer would be equal to $32 \times 32 = 1024$. With the use of the option *Num_Groups*, kernels are separated. Specifically, image and kernels are divided into 32 separate groups. Each which carry out convolutions separately. Without this option, each convolution is calculated with 32 kernels instead of one. This is helpful to generate rules from DICON-$D^2$, since for the time being our rule-extraction algorithm is implemented on CPU and not on GPU.

### 4.2.1. Rule Extraction from VDIMLP

In the upper layers we have an MLP with 512 inputs, 256 hidden neurons and ten output neurons. A VDIMLP approximates this MLP to generate propositional rules. The number of stairs in the staircase activation function was fixed to 50 and 100. Table 3 illustrates the results. Specifically, the first row of this Table is related to the original CNN, while the others provide results obtained by two different VDIMLPs. Columns from left to right designate:

- train accuracy;
- predictive accuracy on the testing set;
- fidelity;
- predictive accuracy of the rules (testing set);
- predictive accuracy of the rules when rules and VDIMLPs agree (testing set);
- number of extracted rules;
- average number of rule antecedents.

**Table 3.** Results obtained by a convolutional neural networks (CNN) and its approximation with a virtual discretized interpretable multi-layer perceptron (VDIMLP) subnetwork in the top layers.

| | Tr. Acc. | Tst. Acc. | Fid. | Rul. Acc. (1) | Rul. Acc. (2) | #Rul. | Avg. #Ant. |
|---|---|---|---|---|---|---|---|
| CNN | 99.55 | 99.39 | - | - | - | - | - |
| VDIMLP ($\Theta = 50$) | 99.45 | 99.31 | 98.16 | 97.68 | 99.44 | 1734 | 11.4 |
| VDIMLP ($\Theta = 100$) | 99.45 | 99.36 | 98.27 | 97.82 | 99.47 | 1570 | 11.6 |

Fidelity of the two VDIMLPs is above 98%, which is high. Furthermore, the predictive accuracy of the rules when the rules and the VDIMLPs agree is a bit above that provided by the original CNN.

Finally, it is worth noting that by increasing the number of stairs, performance in terms of predictive accuracy and fidelity increases slightly. This is not surprising, since the approximation of MLP by VDIMLP is better.

In [19], we tackled the MNIST classification problem with small ensembles of DIMLPs trained by arcing [38]. To accelerate the execution time, we also reduced the size of the images from $28 \times 28$ to $11 \times 11$. Table 4 depicts new results obtained by an ensemble of three DIMLPs and C4.5 DTs. The predictive accuracy obtained by DIMLP ensembles is substantially higher than that provided by C4.5 (98.0% versus 90.1%), but the predictive accuracy of the rules is quite close (89.6% versus 89.2%). Rules extracted from DIMLP ensembles are very numerous compared to those generated from C4.5, but it is difficult to compare their numbers, because the former are unordered, and the latter are ordered.

**Table 4.** Results obtained by a DIMLP ensemble and C4.5 decision trees (DTs). The first and the second row takes into account $11 \times 11$ images, while the third is related to $28 \times 28$ images.

|                          | Tr. Acc. | Tst. Acc. | Fid. | Rul. Acc. (1) | Rul. Acc. (2) | #Rul. | Avg. #Ant. |
| ------------------------ | -------- | --------- | ---- | ------------- | ------------- | ----- | ---------- |
| DIMLP-ens ($11 \times 11$) | 99.7     | 98.0      | 90.7 | 89.6          | 98.7          | 7144  | 9.1        |
| C4.5 ($11 \times 11$)      | 97.9     | 90.1      | -    | 89.2          | -             | 452   | 9.9        |
| C4.5 ($28 \times 28$)      | 97.7     | 89.3      | -    | 88.4          | -             | 392   | 10.6       |

### 4.2.2. Rule Extraction from DICON-$D^2$

Each rule generated from VDIMLP gives rise to a set of rules in the convolutional layers, some of which are illustrated here. For clarity, we visualize a propositional rule with its centroid. Specifically, for all its covered samples we calculate every pixel average value. Then, we symbolize rule antecedents by colored dots; green dots represent antecedents given as $a_i > t_i$, $a_i$ being a rule antecedent and $t_i \in I\!R$. Similarly, red dots indicate antecedents given as $a_j \leq t_j$. Note also that a rule antecedent, which is both red and green is colored in yellow.

To illustrate at the global level examples of rules with their centroids and their discriminant pixels, we chose the most activated rule from VDIMLP ($R_1$) and three randomly selected rules. Figure 4 shows 16 rules generated from rule $R_1$. $R_1$ covers 2479 training samples and involves 157 rules in the convolutional layers. We rank rules produced by VDIMLP or DICON-$D^2$, according to the number of covered samples. For instance, the centroid illustrated in the top left of the Figure 4 covers 107 training samples, while the rule on the bottom right is related to 80 training samples. As can be seen, these centroids present slightly different orientations of "1". In addition, for each centroid, the majority of antecedents is red and lie very often near the edge of the number.

Another example of generated centroids is shown in Figure 5 for number "2". The rules shown were generated from rule $R_{131}$ in the upper layers, covering 608 samples in the training set. The total number of rules produced by DICON-$D^2$ in the convolutional layers is 124. Please note that the centroid on the top left of the Figure covers 35 training samples. Again, most rule antecedents are red and lie near the edge. The majority of red dots are in the hollow between the lower and upper part of the number. Moreover, green dots are within the number and colored dots are absent above or below the number. For the classification, the neural network mainly takes into account a small surface within this hollow, which is an important feature of the figure.

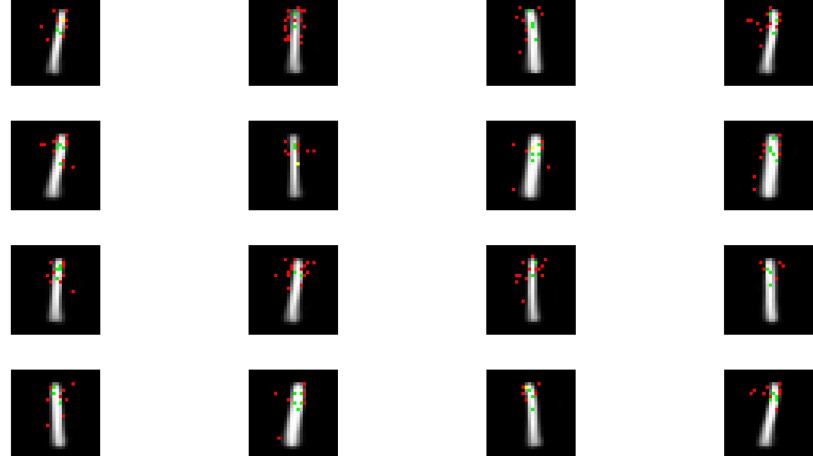

**Figure 4.** Centroids of 16 rules with their antecedents generated from $R_1$ in the fully connected layers.

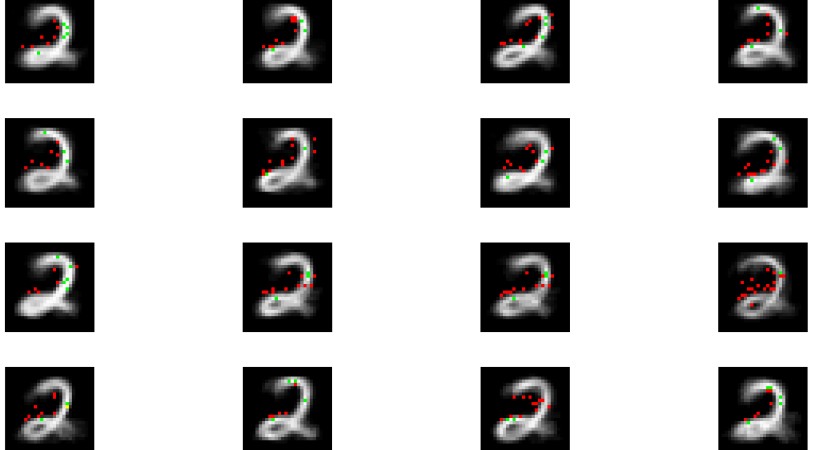

**Figure 5.** Centroids of 16 rules with their antecedents generated from $R_{131}$ in the fully connected layers.

Figure 6 illustrates several centroids for number "7" related to rule $R_{67}$ in the fully connected layers. This rule covers 873 training samples, the total number of rules generated in the convolutional layers being equal to 204. The centroid on the upper left of the Figure covers 18 training samples. For almost all these centroids, we notice that on the upper left of the number there is at least one green antecedent with several red antecedents near the edges.

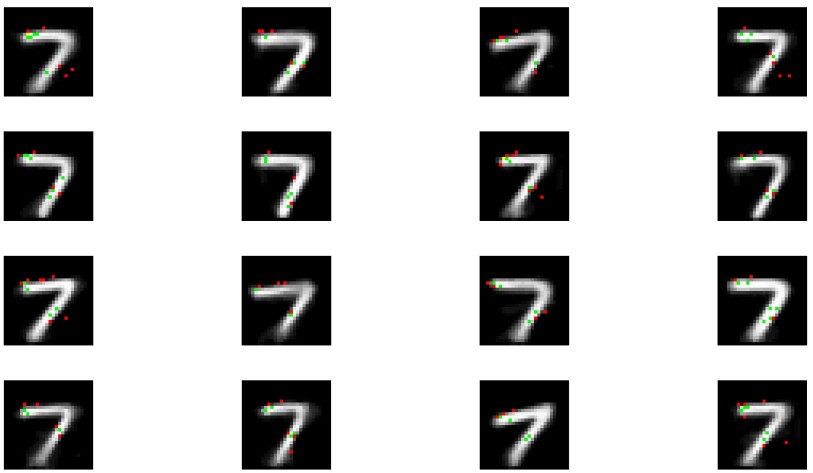

**Figure 6.** Centroids of 16 rules with their antecedents generated from $R_{67}$ in the fully connected layers.

Figure 7 depicts several centroids related to number "8". The rule that generates these centroids is $R_{168}$ in the fully connected layers. It covers 496 training samples and it gives rise to 218 rules in the convolutional layers. It can be noticed that in all the lower hole of digit "8" lie several red dots.

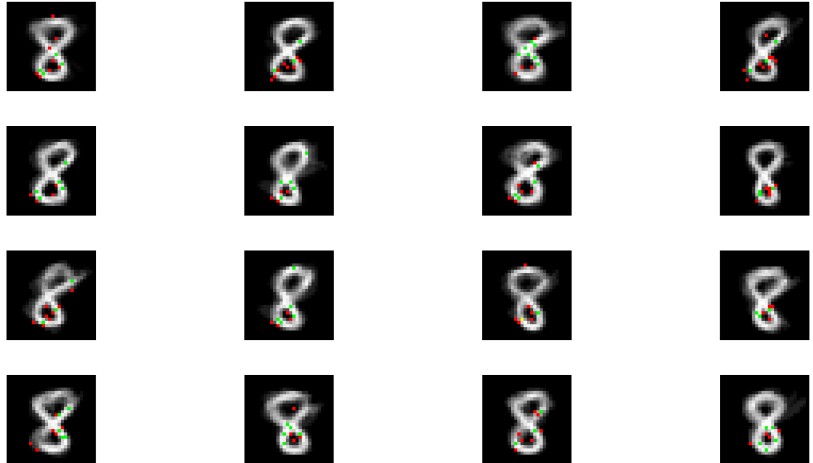

**Figure 7.** Centroids of 16 rules with their antecedents generated from $R_{168}$ in the fully connected layers.

It is clear from the examples illustrated above that we can explain CNN classifications. In fact, the extracted rules emphasize several discriminatory points in particular regions of the pictures, such as:

- the top of "1";
- the main hollow between the lower and upper part of "2";
- the top left part of "7";
- the lower hole of "8".

These particular sets of points could be considered to be relevant features for the classification. Moreover, threshold values of antecedents precisely indicate the conditions under which a classification is performed. Since it is not always possible to make a correct discrimination with only one feature, several features could also be combined. For instance, with number "8" several centroids include a discriminant pixel in the upper hole in addition to the lower hole. Therefore, for a given number, an explanation could resemble to a sentence like: "This number is classified as eight, because it presents several discriminating pixels in the lower hole, and at least a discriminating pixel in the top hole". It should be clear in the mind of the reader that the rule-extraction algorithm determines only the discriminating pixels, which represent low-level features. High-level features such as "the top left part of the number" are strictly related to the classification problem. With respect to medical images, the assistance of specialists would certainly be helpful in targeting specific diagnostic features.

*4.3. Skin-Cancer Dataset*

This skin-cancer classification problem is based on $224 \times 224$ colored images. The dataset includes a training set of 2637 samples and a test set of 660 samples. The classification targets are "benign" and "malignant".The former class represents 54.6% of the training set and 54.5% of the testing set. A tuning set randomly selected from the training set consisting of 1000 samples was used for early stopping. To accelerate the execution time of the rule-extraction algorithm, we decided to transform the $224 \times 224$ colored images into $28 \times 28$ gray images, by averaging $8 \times 8$ pixel blocks. Performing the experiment with lower resolution images may miss something that is present in the original images. Note, however, that interesting elements might remain visible.

### 4.3.1. Rule Extraction with 28 × 28 Images

We defined the same CNN architecture as the one used for the MNIST dataset. (cf. Section 4.2). We conducted ten training trials with data augmentation, due to the small size of the training set. Specifically, with the Keras library [39] it is possible to apply small translations and small deformations to the original training samples. As a baseline we also included C4.5 decision trees from which ordered rules can be generated [33] and ensembles of 25 DIMLPs trained by bagging [32]. Please note that these two models learned without data augmentation. The averaged results are presented in Table 5.

**Table 5.** Results obtained by CNNs, decision trees and ensembles of DIMLPs on the skin-cancer dataset with 28 × 28 images. The first row is related to the results with data augmentation during training.

|  | Tr. Acc. | Tst. Acc. | Fid. | Rul. Acc. (1) | Rul. Acc. (2) | #Rul. | Avg. #Ant. |
|---|---|---|---|---|---|---|---|
| CNN (augm. data) | 85.2 (1.2) | 82.0 (1.0) | - | - | - | - | - |
| CNN | 98.9 (0.8) | 81.9 (0.7) | - | - | - | - | - |
| C4.5 | 98.5 (0.2) | 69.8 (1.7) | - | 70.9 (0.7) | - | 26.1 (6.7) | 4.6 (0.6) |
| DIMLP-ens | 80.5 (0.3) | 75.6 (0.2) | 90.2 (1.4) | 73.7 (1.0) | 77.3 (0.5) | 339.2 (30.6) | 4.9 (0.2) |

By large, average predictive accuracy obtained by CNNs is higher than that provided by decision trees and DIMLP ensembles. An intuitive reason is that CNNs are robust to image translations and deformations. Note also that decision trees generate less rules than DIMLP ensembles, because the expression power of ordered rules is stronger than that of unordered rules.

To present several examples of rules generated in the lower layers, we randomly selected one of the ten CNNs. We first extracted rules from the fully connected layers (VDIMLP subnetwork). Table 6 illustrates the characteristics of the generated ruleset.

**Table 6.** Results obtained by a CNN and its approximation with a VDIMLP subnetwork in the top layers.

|  | Tr. Acc. | Tst. Acc. | Fid. | Rul. Acc. (1) | Rul. Acc. (2) | #Rul. | Avg. #Ant. |
|---|---|---|---|---|---|---|---|
| VDIMLP ($\Theta = 50$) | 82.6 | 81.8 | 94.5 | 80.3 | 82.9 | 222 | 6.9 |

Afterward, we generated symbolic rules from the convolutional layers with the use of DICON-$D^2$. We randomly selected from VDIMLP a rule belonging to the "malignant" class and another belonging to the "malignant" class. For instance, the seventh rule extracted from VDIMLP ($R_7$) covers 190 training samples and 48 testing samples, respectively. Its accuracy on the training set is 90%, with 87.5% testing samples correctly classified. From a DICON, with respect to $R_7$ we generated 67 rules belonging to the positive class (e.g., "malignant"). Figure 8 depicts the centroids of nine rules out of 67; each of them covering three or four training samples. It is worth noting that most of the red dots tend to be in the dark areas, while almost all of the green dots are in the light background. The red dots in the dark areas indicate points of discrimination to look at, while in many cases, the green dots delimit light and dark areas.

Figure 9 shows the four training samples that activate the rule illustrated in the upper right-hand corner of Figure 8. Although these four cases are very different, the same rule makes it possible to classify them correctly.

Figure 10 presents three training samples covered by the rule represented in the upper left-hand corner of Figure 8. Contrary to the previous Figure, the three cases are more similar. Please note that the examples on the left and right each activate an additional rule. As a result, the red and green dots are more numerous, because they represent the antecedents of the supplementary rules.

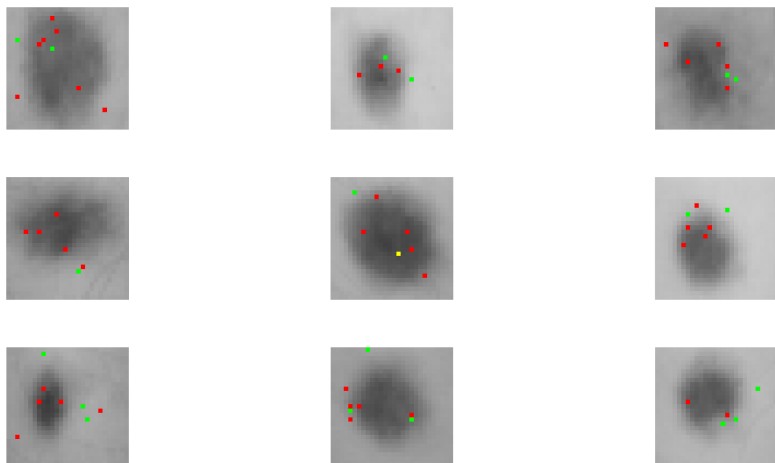

**Figure 8.** Centroids of nine rules with their antecedents generated from $R_7$ in the fully connected layers.

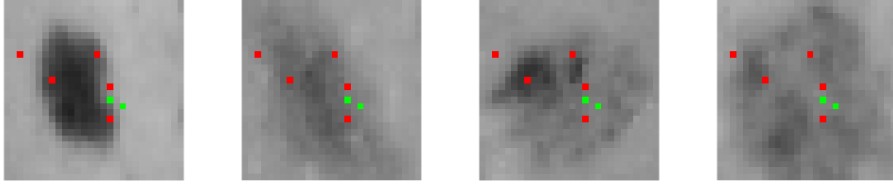

**Figure 9.** An example of four training samples belonging to the "malignant" class, correctly classified by the same rule extracted from a discretized interpretable convolutional network (DICON).



**Figure 10.** Three training samples belonging to the "malignant" class. The example in the middle activates a single rule, while the examples on the left and right each activate an additional rule.

With respect to the sixth rule represented in Figure 8, Figure 11 depicts three training covered samples. Again, the example on the left activates three additional rules, while one additional rule covers the example in the right. Please note that in the example on the left we can clearly see different colored dots in the bright area above the dark area.



**Figure 11.** Three training samples belonging to the "malignant" class. The example in the middle is covered by a single rule, while the examples on the left and right activate four and two rules, respectively.

Figure 12 depicts the centroids of another nine rules extracted from a DICON with respect to $R_7$ generated by VDIMLP, each of them covering three training samples. Figure 13 presents three training samples covered by the second rule, with respect to the upper left-hand of Figure 12. Please note that the second training sample with a higher number of colored dots also activates another rule.

Furthermore, Figures 14 and 15 provide other examples of training samples covered by the penultimate and last centroids of Figure 12.

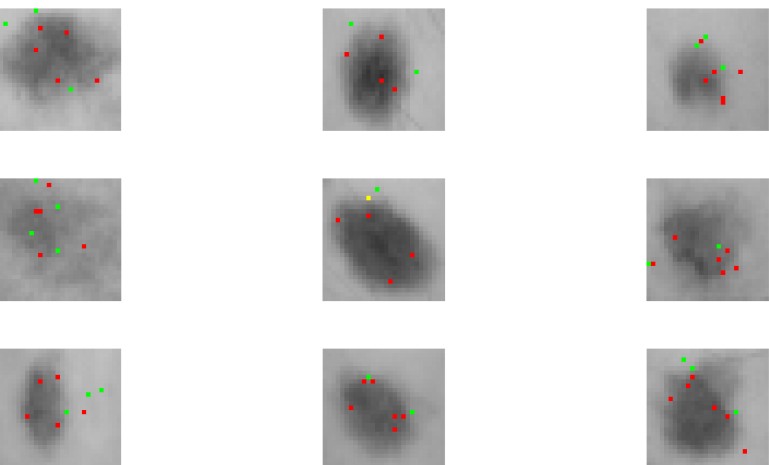

**Figure 12.** Centroids of nine rules with their antecedents generated from $R_7$ in the fully connected layers.

The number of red dots in the main dark area could be an indicator of malignancy (cf. Figures 9–11 and 13–15). Perhaps, in some cases these red dots emphasize non-uniform neighborhoods, such as in Figures 13 and 15. In these same Figures with the addition of Figure 14, the green dots are outside the main dark area. Many of them play the role of delimiters between dark and light areas.



**Figure 13.** Samples belonging to the "malignant" class, sharing a rule. The first and the second case activate a single rule, while the second is covered by two rules.

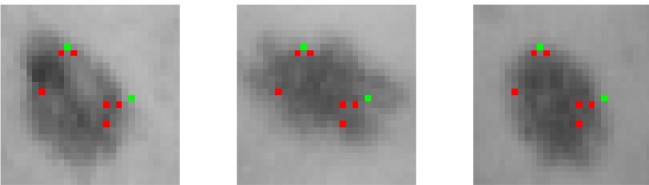

**Figure 14.** Samples belonging to the "malignant" class, activating the same rule.



**Figure 15.** Samples belonging to the "malignant" class, sharing a rule. The first and the third case activate a single rule, while the second is covered by two rules.

As another example of visualization, $R_{17}$ extracted from VDIMLP belongs to the "benign" class. It covers 105 training samples and 31 testing samples, respectively. The accuracy of this rule on the training set is 89.5%, and 90.3% on the testing set, respectively. From a DICON, with respect to $R_{17}$ we

generated 63 rules belonging to the "benign" class. Figure 16 depicts the centroids of nine rules; each of them covering between three and six training samples. Figures 17–20 present the training samples covered by the rule represented in the upper left-hand of Figure 16, the fourth rule (counting from left to right and from the top to the bottom), the sixth rule and the last rule, respectively. In several cases, a training sample activates more than a rule, as observed for $R_7$. It is worth noting that in these rules belonging to the negative class, fewer red dots appear in the dark regions compared to the rules of the "malignant" class.

Apart from the number of red dots in the main dark area, it is difficult to tell what the difference is between the two classes of samples, in general. Nevertheless, it is worth noting low contrast and more diffuse patterns in Figures 18 and 19. A consequence could be the presence of a greater number of green dots than red dots.

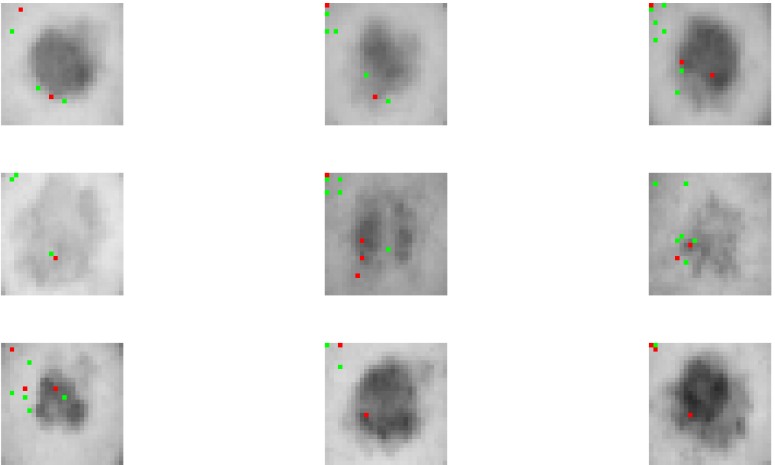

**Figure 16.** Centroids of nine rules with their antecedents generated from $R_{17}$ in the fully connected layers.

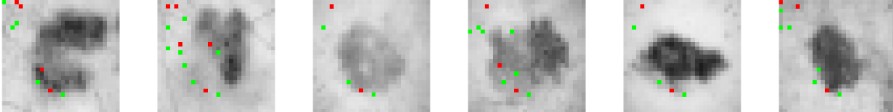

**Figure 17.** Samples belonging to the "benign" class, sharing a rule. The third and the fifth case are covered by a single rule, while the others present a higher number of colored dots related up to four activated rules.

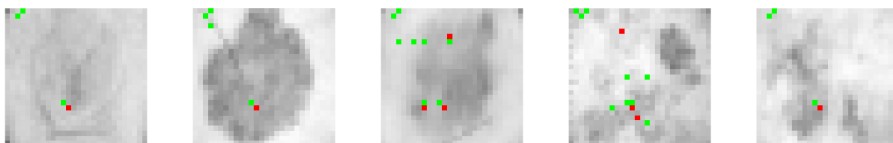

**Figure 18.** Samples belonging to the "benign" class, sharing a rule. The first and the fifth case are covered by a single rule, while the others present a higher number of colored dots related up to three activated rules.

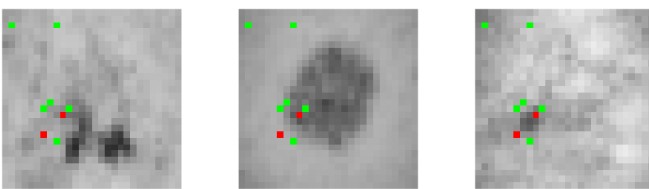

**Figure 19.** Samples belonging to the "benign" class, activating the same rule.

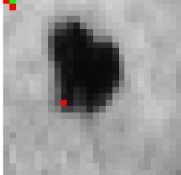 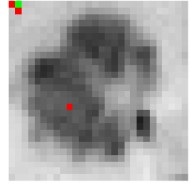 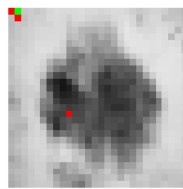

**Figure 20.** Samples belonging to the "benign" class, activating the same rule.

In Figure 20 the top left pixels are picked up by a rule, suggesting that the influence of the corner in the classification of "benign" cases is much higher than one would expect. As an explanation, let us analyze the two histograms represented in Figure 21, relating to the values of the two pixels located in the first row and the first column ($x_{11}$) and the second row and the second column ($x_{22}$). Specifically, each histogram characterizes the distribution of the training samples with respect to class "malignant" (yellow bars) and class "benign" (white bars). The rule antecedents for the two histograms are:

- $x_{11} < 0.75$;
- $x_{22} < 0.77$.

Thus, for the first histogram we clearly observe that below 0.5 a significant difference in the number of samples between the two classes is present. This is also true between 0.6 and 0.75. A similar argument is valid for the second histogram for the values between 0 and 0.5. Regarding the pixel located in the first row and second column, the histogram is not shown but the explanation is alike. These two histograms also tell us that a substantial number of "benign" training samples present gray/dark areas in the top left corner. This particularity could be related to the process of image capture.

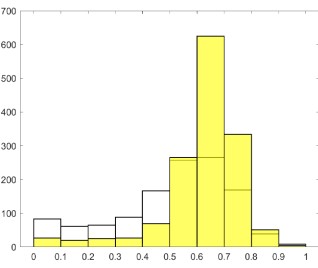 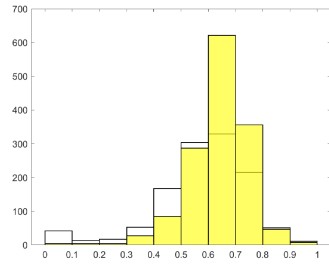

**Figure 21.** Two histograms corresponding to the values of two pixels located in the top left corner. Each histogram characterizes the distribution of the training samples with respect to class "malignant" (yellow bars) and class "benign" (white bars).

Figure 22 illustrates on the left-hand side a case classified "benign", as well as colored dots corresponding to the first rule of an ordered ruleset generated from a decision tree and on the right side, the same case with a rule extracted from a DICON related to $R_{17}$. The rule related to DICON presents less antecedents than those generated from C4.5. We also remark a certain proximity between the yellow dot on the left picture and a green dot in the right picture, which is just offset by two positions. Another similarity lies in the upper left corner, where the DICON and C4.5 both display a red dot at a short distance from each other. A little lower on the right is a green dot, which is just a lower position on the left image. Overall, we have highlighted for this "benign" case three discriminatory points which are very close for the two rule-extraction techniques.

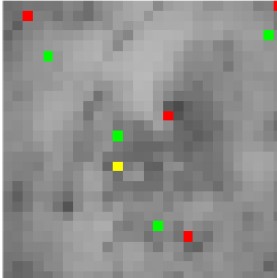 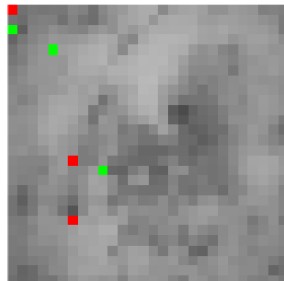

**Figure 22.** A case classified as "benign" by both C4.5 (left) and a CNN (right). Colored dots represent the rule antecedents.

### 4.3.2. Rules Generated by Transfer Learning from VDIMLP

Convolutional pretrained networks can be applied to this classification problem. For instance, VGG was trained with many colored images of size $224 \times 224$, belonging to a thousand of object classes [40]. We provided this skin-cancer dataset to a pretrained VGG, which generates for each sample 512 outputs. Therefore, we obtained a new dataset, which is suitable to train any transparent model. We used VDIMLPs and DIMLP ensembles trained by bagging, since they demonstrated to be very accurate models [41]. As default architectures, VDIMLPs and DIMLP ensembles possessed only a hidden layer, with several neurons equal to the number of inputs. In addition, each ensemble included 25 DIMLPs. Table 7 presents the averaged results obtained after ten training trials.

**Table 7.** Results obtained by transfer learning with a VGG network. The first row is related to VDIMLPs in the upper layers, while the second row concerns ensembles of DIMLPs replacing VDIMLPs.

|  | Tr. Acc. | Tst. Acc. | Fid. | Rul. Acc. (1) | Rul. Acc. (2) | #Rul. | Avg. #Ant. |
|---|---|---|---|---|---|---|---|
| VDIMLP | 86.0 (0.9) | 83.8 (1.2) | 95.0 (0.9) | 83.1 (1.1) | 85.2 (1.2) | 199.1 (11.9) | 5.7 (0.2) |
| DIMLP-ens | 87.1 (0.3) | 84.9 (0.3) | 95.4 (0.7) | 83.9 (0.8) | 86.0 (0.4) | 181.2 (29.0) | 5.8 (0.5) |

To generate propositional rules putting into play antecedents depending on the input layer, we would require a GPU implementation of DICON. Indeed, VGG is a deep convolutional network with 16 layers, which would take too much time to make it work with rule extraction on CPU. At this stage, the rules generated by DIMLP ensembles are valuable for characterizing how samples are clustered, which could help determine how a classification is achieved.

### 4.4. Discussion

In the two classification problems on which we applied our new rule-extraction technique, we obtained for the extracted rules high predictive accuracy. As a result, the difference of accuracy with respect to DTs was substantial: 97.8% versus 89.2% for MNIST; and 80.3% versus 72.1% for skin cancer (which is the best predictive accuracy reached by C4.5 with respect to an average equal to 70.9%). The fidelity of the rules produced by CNNs was also very high, with 98.3% on MNIST and 94.5% on skin cancer. Regarding the number of rules, those produced by CNNs are significantly more numerous than those produced by DTs. However, any ruleset produced by C4.5 is ordered; therefore, many antecedents of rules are implicit. Specifically, in an ordered list of rules, only the first in the list contains all the antecedents. From the second rule in the list, many antecedents are implicit. With the unordered rulesets generated from CNNs, all the antecedents are explicit, and each rule can then be examined independently. Note also that unordered rules must be extracted from VDIMLPs, because if ordered rules with many implicit antecedents were used, rule extraction from DICONs in the convolutional layers would be incomplete.

Rules produced in the upper layers from VDIMLP explain how classification decisions are reached, with respect to filter values. Specifically, the rule antecedents indicate combinations of responses from

the filters in the last convolutional layer. For the MNIST problem, the interactions between the filters appear very complex, since more than 11 filter activations are involved, on average. At this point, each unordered rule forms a cluster represented by a centroid. With the rules produced in the lower layers from DICON-$D^2$, each cluster created by VDIMLP is further divided into new subclusters. Each one of them is again represented by a centroid, but in addition, we know the location of the discriminatory pixels.

Since our rule-extraction algorithm is global, we can inspect the knowledge embedded in a CNN from the clusters produced by VDIMLP and the clusters generated from DICON-$D^2$, with also the discriminant variables. Thus, it is possible to carry out a global analysis. Furthermore, local analysis for a given sample can be achieved by determining all the activated rules with their discriminant variables.

A single rule generated from VDIMLP can result in hundreds of rules at the DICON level. This may seem like a lot, but at the same time, it reflects the complexity of what is processed by the convolutional layers and also the high dimensionality of the classification problem. To remedy this, we could possibly simplify in future work the union of all the rules obtained from the different DICONs for each of the classes.

With the classification of numbers (MNIST dataset), we remarked that very often the discriminant regions are located along the edges, as well as in particular places without signal (as the lower hole of number "8"). A number is very well defined, even if different individuals will write it in a different way. However, for skin-cancer images, the classes are not as clear as with numbers. Thus, we may wonder about the discriminating pixels represented by the rules. Perhaps the red dots in the dark regions simply indicate local anomalies. There could eventually be a particular texture that would be strongly related to this skin disease and perhaps specialized physicians are able to recognize it. Our hope is that they designate in their neighborhood relevant parts of the image.

A new open question is to find out whether it is worth learning the samples covered by each rule produced in the upper layers from the VDIMLPs. Transparent models such as DTs or DIMLP ensembles could accomplish this task. On the contrary, generating rules from DICON could require a very long execution time with deep models. On the one hand, an adverse argument is that a less expressive model will find it difficult to replace the CNN. On the other hand, a favorable argument is that a less powerful model might be able to learn a less complex classification, which is binary.

## 5. Conclusions

We presented a new global method that generates propositional rules from CNNs comprising 2D convolutional layers or 2D max-pooling layers. The key idea behind our rule-extraction algorithm is to approximate a CNN by two subnetworks from which it is possible to generate rules. The first subnetwork approximates the upper layers and the second approximates the lower layers. Rules produced by the upper layers were chained down to those extracted from the lower layers. In such a way, the obtained rules presented in the antecedents the input variables of the datasets. We first applied our rule-extraction technique to images of digits. The visualization of the rules by their centroids with their discriminant features allowed us to distinguish several particular subregions. Secondly, a similar characterization was performed with skin-cancer images. Specifically, by means of several rule examples, we emphasized relevant pixels for image classification. An interesting question will be whether physicians believe that the proposed rule antecedents with their values are relevant to the final diagnosis.

**Author Contributions:** Conceptualization, G.B. and S.F.; methodology, G.B.; software, S.F. and G.B.; validation, G.B. and S.F.; formal analysis, G.B. and S.F.; investigation, G.B. and S.F.; resources, G.B. and S.F.; data curation, G.B. and S.F.; writing—original draft preparation, G.B.; writing—review and editing, G.B.; visualization, G.B.; supervision, G.B.; project administration, G.B. All authors have read and agreed to the published version of the manuscript.

**Funding:** This research received no external funding.

**Conflicts of Interest:** The authors declare no conflict of interest.

## Abbreviations

The following abbreviations are used in this manuscript:

CNN         Convolutional Neural Network
MLP         Multi Layer Perceptron
DIMLP       Discretized Interpretable Multi Layer Perceptron
VDIMLP      Virtual Discretized Interpretable Multi Layer Perceptron
DICON       Discretized Interpretable Convolutional Network
XAI         Explainable Artificial Intelligence
DT          Decision Trees
LIME        Locally Interpretable Model Agnostic Explanations
LORE        Local Rule Based Explanations

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
