# Peer review of "A Two-Step Rule-Extraction Technique for a CNN"

_electronics, doi:10.3390/electronics9060990_

Round 1

Reviewer 1 Report

This is a good attempt to generate rules from CNNs in order to increase explainability of "Black box" models. Nicely explained/detailed manuscript apart from minor missing details. 

Thyroid dataset was used to show that VDIMLP approximates MLP. The other two datasets were used to show the two-step rule extraction. An explanation or note why the Thyroid dataset was not given the same treatment would be helpful for the readers.

To accelerate the execution time, transforming 224 by 224 images to 28 by 28 does not seem to be a great reasoning. Although it is a reason, conducting the experiment with original images may reveal something that is hidden in the smaller resolution images. Adding some sort of comment in this regard would be helpful.

"The average predictive accuracy is equal to 84.9%, which is better than that obtained with the CNN trained on 28x28 images (82%)." This statement is misleading. The comparison is not fair as the CNN is trained on 28 by 28 images while features were obtained using VGG on 224 by 224 images. 

In the Skin data set, Figure 20, it seems that the top left pixels are being picked up by the algorithm, suggesting that the influence of the corner in the classification task is much higher than one would expect. Some explanation would be useful for the reader.

Some minor comments:

Line 89-90: reference 25 is not for Nguyen et al. Please check and correct it.

Line 92: highly dimensional problems should be changed to high-dimensional problems.

Figure 1 caption: please add a bit more detail in the caption.

Line 417 and 418: both of them start with "On the other hand". Please change one of them.

Author Response

Reviewer:

1-Thyroid dataset was used to show that VDIMLP approximates MLP. The other two datasets were used to show the two-step rule extraction. An explanation or note why the Thyroid dataset was not given the same treatment would be helpful for the readers.

Our answer:

We added an explanation at the beginning of the Results section:
"Note that the input features of the Thyroid classification problem are not related to images or temporal series. Thus, at priori it is more appropriate to learn this dataset with a model such as an MLP, rather than a CNN."

Reviewer:

2- To accelerate the execution time, transforming 224 by 224 images to 28 by 28 does not seem to be a great reasoning. Although it is a reason, conducting the experiment with original images may reveal something that is hidden in the smaller resolution images. Adding some sort of comment in this regard would be helpful.

Our answer:

We added at the beginning of Sect. 4.3: "Performing the experiment with lower resolution images may miss something that is present in the original images. Note, however, that interesting elements might still remain visible"

Reviewer:

3- The average predictive accuracy is equal to 84.9%, which is better than that obtained with the CNN trained on 28x28 images (82%)." This statement is misleading. The comparison is not fair as the CNN is trained on 28 by 28 images while features were obtained using VGG on 224 by 224 images.

Our answer:

Since the comparison is not fair, we removed this sentence.

Reviewer:

4- In the Skin data set, Figure 20, it seems that the top left pixels are being picked up by the algorithm, suggesting that the influence of the corner in the classification task is much higher than one would expect. Some explanation would be useful for the reader.

Our answer:

We added an explanation related to a new figure showing two histograms of two pixels in the top left.
The main reason for the influence of the corner is that a substantial number of "Benign" training samples present dark/gray areas in the top left corner (see text and figure 21).

-----------------------------------------------------------------------------------------

MINOR

Line 89-90: reference 25 is not for Nguyen et al. Please check and correct it.

Yes, indeed this reference has been corrected.

Line 92: highly dimensional problems should be changed to high-dimensional problems.

This has been changed.

Figure 1 caption: please add a bit more detail in the caption.

Yes, we added more details in the caption of the figure (see in the text).

Line 417 and 418: both of them start with "On the other hand". Please change one of them.

We checked in the text. At 417 the sentence starts with: "On the one hand ...". At 418 the sentence starts with: "On the other hand ..." (which is different).
Thus, since they are different we did not change anything.

Reviewer 2 Report

The authors propose a two-step technique that generates propositional rules from CNN in order to explain the decisions provided by the model.

It is a really interesting contribution.

Some minor comments:

It would be interesting to add info about VDIMLP results in table 5 for an easy comparison.

On skin cancer you have used C4.5 Decision Trees and also DIMLP-ens method. It would be nice to see how these methods perform in the other 2 datasets. Results are nice but it would be better if they are compared with other existing methods like those.

Author Response

Reviewer:

1- It would be interesting to add info about VDIMLP results in table 5 for an easy comparison.

Our answer:

For this Table we added new results about VDIMLP. In this revised version, the old Table 5 is now Table 7.

Reviewer:

2- On skin cancer you have used C4.5 Decision Trees and also DIMLP-ens method. It would be nice to see how these methods perform in the other 2 datasets. Results are nice but it would be better if they are compared with other existing methods like those.

Our answer:

We added new results about the Thyroid dataset. Experiments were performed with decision trees and DIMLP ensembles (see Table 2 and the text close to it).
Idem for the Mnist dataset (see Table 4 and the text close to it).

Reviewer 3 Report

1) The study will be completed if the authors justify the proposed Architecture: What is the main advantage of combining VDIMLP and CNN ? 

2) I didn't find a comparison between the proposed architecture performance with Decision trees and DIMLP for the same datasets. 

Author Response

Reviewer:

1) The study will be completed if the authors justify the proposed Architecture: What is the main advantage of combining VDIMLP and CNN ?

Our answer:

We added in the last paragraph of the introduction:
"The main advantage of introducing VDIMLP in the CNN is that the hard problem of rule extraction is divided into many rule extraction subproblems that are easier to solve. These subproblems depend on the rules generated by VDIMLP in the upper layers. Since these rules are unordered, each rule is itself a classifier that allows us to decompose the original classification problem into many binary classification problems. Each binary classification problem is approximated by a DICON in the lower layers of the CNN, which is only a small part of it. Practically, rule extraction from DICONs run in parallel, which is also a clear advantage."

Reviewer:

2) I didn't find a comparison between the proposed architecture performance with Decision trees and DIMLP for the same datasets.

Our answer:

Yes, the reviewer is right. For the Skin-Cancer dataset CNNs were trained with data augmentation, whereas DIMLP-ens and C4.5 decision trees were trained without data augmentation. As a remedy, we performed new experiments using CNNs without data augmentation. We added the new results in Table 5. Note that the new average predictive accuracy is very similar to the one obtained with data augmentation.

Reviewer 4 Report

The paper presents a rule extraction technique for CNN. It includes the background and how the proposed method is different from existing papers. However, the result section must be improved:

  1. Explain what is option Num_Groups in line 265.
  2. Mathematic explanation of how the predictive accuracy of rules are calculated is missing.
  3. How the rules were selected for representation purposes?
  4. The rules are not covering the details of images, e.g. Fig. 5. How they can result in very high prediction.
  5.  How the rules can explain CNN in line 313 need more explanation.
  6. It is important to compare the visualisation of various rule extraction techniques for 1-2 examples.
  7. There are many figures for the cancer dataset without enough discussion and most of them look very similar. Authors can focus more on discussion and comparison than providing lots of similar figures.
  8. Discussion section must be extended to include more discussion around the generated results and not only the future work and limitations.

Author Response

Reviewer:

1- Explain what is option Num_Groups in line 265.

Our answer:

As an explanation we added in the text:

"Specifically, image and kernels are divided into 32 separate groups. Each which carry out convolutions separately. Without this option, each convolution is calculated with 32 kernels instead of one."

Reviewer:

2- Mathematic explanation of how the predictive accuracy of rules are calculated is missing.

Our answer:

We added in the text at the beginning of the "Result" Section a formula that makes it possible to calculate the accuracy of a ruleset. We describe the different cases corresponding to ordered/unordered rulesets (see equation (8)).

Reviewer:

3- How the rules were selected for representation purposes?

Our answer:

The purpose was to both illustrate examples of rules at the global level with their centroids, as well as their discriminant pixels. For the MNIST dataset the purpose was first to show centroids of the most activated rule, with respect to VDIMLP (see Fig. 4). The other three figures representing three other rules were randomly selected. For the Skin-Cancer classification problem, we both depicted centroids and particular cases with their discriminant pixels. The choice of R7 and R17 with respect to VDIMLP was random. However, our intention was to describe at least a group of rules belonging to the malignant class and another group belonging to the benign class.

We added at the beginning of Sect. 4.2.2: "To illustrate examples of rules at the global level with their centroids and their discriminant pixels, we chose the most activated rule from VDIMLP R_1 and three randomly selected rules."
We also added a sentence in Sect. xxx: "We randomly selected from VDIMLP a rule belonging to the ``Malignant'' class and another belonging to the ``Malignant'' class. "

Reviewer:

4- The rules are not covering the details of images, e.g. Fig. 5. How they can result in very high prediction.

Our answer:

With respect to Figure 5, as we wrote in the text, the majority of rule antecedents are red and lie near the edge. The majority of red dots are located in the hollow between the lower and upper part of the number. Moreover, green dots are within the number and colored dots are absent above or below the number. For the classification, the neural network mainly takes into account a small surface within this hollow, which is an important feature of the figure.
We added this text written above close to this Figure.

The accuracy of the rule is high, because the feature represented by the hollow between the lower and upper part of number "two" is intuitively a relevant feature of ths number.

Reviewer:

5- How the rules can explain CNN in line 313 need more explanation.

Our answer: We added more explanations at the end of Sect. 4.2.2 (see text in bold).

Reviewer:

6- It is important to compare the visualisation of various rule extraction techniques for 1-2 examples.

Our answer:

With an example of a benign case (Fig. 22), we compared C4.5, DICON and their activated rules. See text in bold close to Fig. 22.

Reviewer:

7- There are many figures for the cancer dataset without enough discussion and most of them look very similar. Authors can focus more on discussion and comparison than providing lots of similar figures.

Our answer:

Yes, it is right that for Figures 13, 14, 15, 16, 17, 18, 19, and 20 there are no specific comments, because we thought that the explanations related to Figures 8, 9, 10, and 11 would be sufficient. Nevertheless, we added new text below Fig.12 (in bold) and also after Figure 15. Moreover, a new paragraph has been written after Figure 20.

Reviewer:

8- Discussion section must be extended to include more discussion around the generated results and not only the future work and limitations.

Our answer:

We added a new paragraph to the Discussion Section (the first in bold). It is related to the obtained results.

Round 2

Reviewer 4 Report

Authors answered all my comments.